# Trends in Overall Survival among Patients Treated for Sarcoma at a Large Tertiary Cancer Center between 1986 and 2014

**DOI:** 10.3390/cancers15020514

**Published:** 2023-01-14

**Authors:** Erik Stricker, Damon R. Reed, Matthew B. Schabath, Pagna Sok, Michael E. Scheurer, Philip J. Lupo

**Affiliations:** 1Department of Molecular Virology and Microbiology, Baylor College of Medicine, Houston, TX 77030, USA; 2H. Lee Moffitt Cancer Center, Adolescent and Young Adult Program, Tampa, FL 33612, USA; 3H. Lee Moffitt Cancer Center, Departments of Cancer Epidemiology and Thoracic Oncology, Tampa, FL 33612, USA; 4Section of Hematology-Oncology, Department of Pediatrics, Baylor College of Medicine, Houston, TX 77030, USA

**Keywords:** sarcoma, survival, soft tissue sarcoma, bone sarcoma, epidemiology, cancer registry

## Abstract

**Simple Summary:**

Sarcomas are comparatively rare cancers; thus, large sarcoma studies covering extended time periods are lacking. Therefore, our study analyzed data from 2570 adolescents (15–39 years) and adults (≥40 years) treated at the Moffitt Cancer Center between 1986 and 2014. We aimed to evaluate the impact of characteristics such as sex, age, ethnicity, race, tobacco use, diagnosis, cancer metastasis, treatment, and family history on overall survival among individuals diagnosed with soft tissue or bone sarcomas. The collected data gave us the advantage of including a large patient number and made possible the evaluation of several sarcoma subtypes. Lastly, data collected over 28 years allowed us to look for changes over time, often not possible in small studies and capture improvements in treatment. Our study showed poorer overall survival rates in older adults (≥40 years), current smokers, patients with metastatic cancer, and patients not receiving first-line surgery treatment. There was a moderate improvement in overall survival rates over time, with gastrointestinal stromal tumors experiencing better overall survival in more recent years. We believe that our study provides important findings for the field of sarcoma research and highlights the need for future research to better understand barriers to survivorship.

**Abstract:**

Sarcomas are relatively rare malignancies accounting for about 1% of all cancer diagnoses. Studies on sarcomas comprising large cohorts covering extended time periods are lacking. Therefore, this study aimed to evaluate the impact of demographic, behavioral, and clinical characteristics on overall survival (OS) among individuals diagnosed with soft tissue sarcoma (STS) or bone sarcoma at the Moffitt Cancer Center between 1986 and 2014. Unadjusted and multivariable Cox proportional hazard regression (CPHR) models were constructed to generate hazard ratios (HRs) and 95% confidence intervals (CIs) to evaluate associations between a range of demographic, behavioral, and clinical characteristics, and OS. Additionally, Kaplan–Meier survival curves, associated log-rank statistics, and adjusted CPHR models were generated by time periods based on the year of first contact (1986–1994, 1995–1999, 2000–2005, 2006–2010, 2011–2014) to evaluate for temporal differences in OS. Of the 2570 patients, 2037 were diagnosed with STS, whereas 533 were diagnosed with bone sarcoma. At the time of analysis, 50% of the population were alive. In multivariable analyses, we observed poorer survival for patients ≥ 40 years of age (HR = 1.54, 95% CI = 1.34–1.78), current smokers (HR = 1.18, 95% CI = 1.01–1.37), patients with metastasis (HR = 2.19, 95% CI = 1.95–2.47), and patients not receiving first-line surgery treatment (HR = 2.11, 95% CI = 1.82–2.45). We discovered limited improvements in OS over time among individuals diagnosed with STS or bone sarcomas with the exception of gastrointestinal stromal tumors (GIST), which showed a significant improvement in OS across time periods (*p* = 0.0034). Overall, we identified well-established characteristics associated with OS (e.g., metastasis) in addition to factors (e.g., smoking status) not previously reported to impact OS. Improvements in survival over time have been relatively modest, suggesting the need for improved therapeutic options, especially for those diagnosed with less frequent sarcomas.

## 1. Introduction

Sarcomas are comparatively rare cancers of mesenchymal origin with 16,730 estimated new cases in the United States for 2020, accounting for about 1% of all cancer diagnoses [1]. While these malignancies are relatively heterogeneous, they can be broadly categorized into two groups, soft tissue sarcomas (STSs) and bone sarcomas.

STSs comprise a large variety of subtypes with different histological features and clinical behaviors [2]. While the incidence of STS increases significantly with age [2], these tumors make up a higher proportion of cancers in children (7%) compared to adults (i.e., 5% for <40 years, 3% for <50 years). STSs can be classified into over 50 histological subtypes, with liposarcoma, leiomyosarcomas, and undifferentiated sarcomas as the predominant adult subtypes, and rhabdomyosarcomas as the most frequent pediatric STS [3]. Survival is generally guided by metastasis, tumor grade, size, and depth [4]. In the treatment of STSs, complete surgical removal of the tumor has been shown to be critical in many subtypes, while the effectiveness of chemotherapy and radiation therapy depends on the histological subtype [2]. Furthermore, there have been few therapeutic advances in recent decades outside of gastrointestinal stromal tumors (GIST) [5,6].

Bone sarcomas are predominantly highly metastatic malignancies; however, there are also less frequent bone sarcomas with lower metastatic potential [2,7,8]. Notably, bone sarcomas are largely diagnosed in adolescents and young adults (AYAs) and often require multimodal therapy including intensive chemotherapy, surgery, and/or radiation therapy [2,9]. While osteosarcoma and Ewing sarcoma are more common in children and AYAs, the predominant bone sarcoma in older adults is chondrosarcoma [2]. Prognosis for those diagnosed with bone sarcomas is largely dependent on metastasis, age, size, and site [4]. As with STSs, there have been few advances in therapy since the 1970s [2,10].

An evaluation of historical trends in cancer treatment and survival is important to determine the overall progress toward patient outcomes and to reveal where improvements are needed. In spite of this, there have been limited studies of the epidemiology of sarcomas. Therefore, the goal of this study was to evaluate the impact of demographic, behavioral, and clinical characteristics on overall survival (OS) among individuals diagnosed with sarcoma over several decades in a large institutional cohort.

## 2. Materials and Methods

### 2.1. Study Population

This analysis included 2570 patients with STSs or bone sarcomas who were treated at Moffitt Cancer Center (MCC) between 1986 and 2014. The catchment area of the MCC included 23 counties and with 9.8 million people, nearly 47% of Florida’s population at the time of this study, covering a higher number of African American and minority residents. The study population was selected from a larger patient cohort of 2663 individuals by excluding patients diagnosed at <15 years of age. We further divided the population into five groups based on year of diagnosis: 1. 1986–1994 (N = 207); 2. 1995–1999 (N = 208); 3. 2000–2005 (N = 636); 4. 2006–2010 (N = 795); and 5. 2011–2014 (N = 724). These time periods were selected to obtain comparable time frames and cohort sizes allowing the evaluation of changes in demographics and overall survival within and across decades [10,11].

### 2.2. MCC Cancer Registry Data

The primary data source for this assessment was the MCC Cancer Registry, which includes information from patient electronic medical records on demographics, diagnosis, metastasis, history of smoking, alcohol use, treatment, and other clinical information. A waiver of informed consent was provided for this study by Advarra, Inc. IRB due to its retrospective nature and the size of the dataset. Patients were followed up annually through passive and active methods [11]. The “first course of treatment” was defined by the Cancer Registry as all methods of treatment recorded in the treatment plan and administered to the patient before disease progression, recurrence, or death. If multiple treatments, such as surgery, radiation, or chemotherapy, were part of the initial treatment plan, they were treated as a single unit consisting of multiple treatment modalities. Additionally, in our statistical analysis, we evaluated differences in outcomes between patients with surgery, radiation, or chemotherapy as the only or as part of their initial treatment plan and patients without the corresponding modality in their initial treatment plan. For this analysis, patients were further categorized by age of diagnosis: AYAs (15–39 years) and older adults (≥40 years) according to the definitions by the National Cancer Institute [12]. Tobacco was categorized as self-reported current, former, or never according to the Florida Cancer Data System (FCDS) and cigarette numbers. Where missing, the tumor node metastasis (TNM)/Collaborative Staging (CS) mixed stage recorded was supplemented by pathological TNM staging, staging documented at first contact according to the Florida Cancer Data System (FCDS), and stage summaries by reporting physicians. Metastasis subcategories were combined into localized (stages 1, 2, 3) and metastatic (stage 4). All sarcoma diagnoses were classified as either STSs or bone sarcomas according to the ICD-O-3 classification of histology and behavior (Appendix A). The 43 different ICD-O-3 histology and behavior codes were further grouped into 13 histological subtypes (Appendix A). Due to the low number of patients in the respective groups, epithelioid sarcoma, myxosarcoma, malignant rhabdoid tumor, clear cell sarcoma NOS (except for kidney), giant cell tumor of malignant bone sarcoma, malignant peripheral nerve sheath tumor, and alveolar soft tissue sarcoma were summarized in the “other” histological subtype.

### 2.3. Statistical Analysis

Demographic, behavioral, and clinical characteristics were evaluated using descriptive statistics (counts and proportions). Unadjusted Cox proportional hazard regression models were constructed to calculate the hazard ratio (HR) and 95% confidence interval (CI) for each variable on overall survival (OS). Variables that met the statistical significance threshold or were clinically relevant were included in the multivariable Cox proportional hazard regression models. For the analyses of time periods, all variables that remained significant in the multivariable analyses were included. In addition, Kaplan–Meier survival curves, associated log-rank statistics, and adjusted Cox proportional hazard regression models were constructed, stratifying by time periods. For this study, OS was right-censored at 10 years. All statistical analyses were performed using R version 4.0.2 (R Project for Statistical Computing, www.rproject.org (accessed on 27 June 2020)), and statistical significance threshold was defined at *p* < 0.05 unless otherwise stated.

## 3. Results

Between 1986 and 2014, 2570 patients were diagnosed and treated at the MCC, of which 2037 (79.3%) and 533 (20.7%) were diagnosed with STS and bone sarcoma, respectively. At the time of this evaluation, 50% of the population were alive. Table 1 presents the distributions of the demographic, behavioral, and clinical characteristics of the study population and the unadjusted HRs and 95% CIs of OS. Overall, 50.5% were male, 75.8% were diagnosed at or over 40 years of age, and 89.3% and 86.7% were of non-Hispanic ethnicity and White race, respectively. Based on the unadjusted analysis, factors associated with worse OS included age at diagnosis (≥40 years vs. 15–39 years: HR = 1.28, 95% CI = 1.13–1.46), smoking status (current users vs. never users: HR = 1.24, 95% CI = 1.06–1.44), several clinical variables including metastasis (metastatic vs. localized: HR = 2.45, 95% CI = 2.19–2.74) and first course of therapy that included surgery (no vs. yes: HR = 1.85, 95% CI = 1.65–2.07).

From the multivariable model (Table 2), age at diagnosis of (≥40 years vs. 15–39 years: HR = 1.54, 95% CI = 1.34–1.78), smoking status (current users vs. never users: HR = 1.18; 95% CI = 1.01–1.37), and first course of therapy that included surgery (no vs. yes: HR = 2.11, 95% CI = 1.82–2.45) were still observed in association with worse OS. While OS was not statistically different by sarcoma types (STSs vs. bone, *p* = 0.24), there were observed differences in OS by histological subtype (Appendix A).

Accordingly, we assessed changes in OS survival over time for all histological subtypes separately. We generated Kaplan–Meier survival curves by years of diagnosis and determined statistical differences in 1-year, 5-year, 10-year, and overall survival for STSs, bone sarcomas, and each of the 13 histological subtypes separately (data not shown). The Kaplan–Meier survival curves by years of diagnosis (Figure 1a,b) demonstrated limited improvements in OS even among patients diagnosed in more recent years for STSs (*p* = 0.19) and bone sarcomas (*p* = 0.16). For STSs, the 5-year survival was significantly increased from 45% in 1986–1994 to 64% in 2011–2014 (*p* = 0.0046), while no significant changes could be observed for bone sarcomas. Evaluations by specific histological subtypes demonstrated no significant improvements in OS; see osteosarcoma (*p* = 0.49, Figure 1c), apart from GIST, an increasingly molecularly understood and targetable STS (Figure 1d). We observed a significant increase in OS of patients who were diagnosed with GIST from 2006 and onward (*p* = 0.0034) as well as improved 1-year (*p* = 0.0086) and 5-year (*p* ≤ 0.0001) survival rates.

Finally, we evaluated variables associated with OS split by years of diagnosis. Overall, most variables did not differ in their impact across time periods, but there were some exceptions (Table 3). For example, the impact of tobacco use on OS appeared to vary across time. Specifically, the strongest difference in OS between never and current smokers was observed for 2000–2005 (HR = 1.39, 95% CI 1.06–1.83), whereas in all other time periods (Appendix A), never smokers displayed increased OS when compared to 1986–1994 (1995–1999 HR = 0.67, 95% CI 0.49–0.91; 2006–2010 HR = 0.76, 95% CI 0.59–0.97; 2011–2014 HR = 0.58, 95% CI 0.42–0.79). While adolescents demonstrated better OS compared to adults across all time periods (Table 3), OS for patients with sarcoma was shown to be improved regardless of age at diagnosis (Appendix A). Patients undergoing surgery or not requiring chemotherapy as part of the initial treatment had more favorable outcomes. Interestingly, although surgeries significantly progressed in their impact on OS over time, advances in OS through improved chemotherapy were undetectable in our analysis. In agreement with the result from the Kaplan–Meier survival curves, there were no persistent improvements in OS over time for specific histological subtypes (relative to osteosarcoma) except for GIST and spindle cell sarcoma/NOS (Appendix A).

## 4. Discussion

In a large study of patients diagnosed with STS and bone sarcoma spanning three decades, we observed limited improvements in overall survival, which is consistent with previous reports (particularly for bone sarcomas), indicating a limited progress in treatment outcomes since the 1990s for those diagnosed with sarcomas relative to other malignancies [13]. Nonetheless, there was a modest improvement in OS for those diagnosed with STS (Figure 1b). Specifically, the 5-year survival was significantly increased from 45% in 1986–1994 to 64% in 2011–2014 (*p* = 0.0046). This finding is also consistent with data from the National Cancer Institute’s Surveillance, Epidemiology, and End Results (SEER) program reported by Jacobs et al. (2015) [14].

Consistent with that report, we observed an improvement in OS among STSs relative to bone sarcomas. This may largely be driven by the introduction of imatinib and other kinase inhibitors for the treatment of GIST in 2002 [13,15], a subtype of STS that was present among our study population. GISTs themselves could only be reliably distinguished from other histopathological subtypes with the discovery of gain-of-function mutations in the c-KIT proto-oncogene in 1998, which is why our evaluation of GIST included fewer time periods [16].

To ensure that advances in OS for GIST did not mask overall improvements outside of GIST, we analyzed the differences in survival for each histological subtype separately with Kaplan–Meier survival curves and a multivariate regression model. We could only observe limited improvements for histological subtypes other than GIST which did not consistently exceed the statical significance threshold. Furthermore, differences observed for specific time periods and histological subtypes, such as Ewing sarcoma, leiomyosarcoma, and endometrial stromal sarcoma (Appendix A), are in part debatable due to low patient numbers, in particular in early time periods. We acknowledge that most histological subtypes are very different in nature which warrants the notion of overall separate analysis; however, general results for STSs and bone sarcomas reflected the limited improvements which were observed upon single analysis of each histological subtype but provided a larger study cohort.

Other incremental increases in OS observed for sarcomas in general could be indicative of improvements in supportive care, improvements in selecting patients for particular therapies, earlier detection of metastases, and additional lines and modalities of available therapies [2,9]. Other studies and clinical trials confirmed that improved multidisciplinary care and increased access to effective systemic therapies have significantly increased OS for certain sarcoma subtypes [17,18,19,20,21]. Moreover, there were additional factors associated with OS including age at first contact, smoking status, metastatic disease, and certain treatment strategies. These factors had also been reported in previous studies [22,23].

It is widely accepted that metastatic and non-operated patients have poorer survival than patients with local tumors or those who undergo surgery. Accordingly, a study using data from SEER evaluating outcomes among those diagnosed with sarcoma between 2002–2014 reported improved survival for those treated with surgery as part of first-line therapy [24]. With surgery being the only treatment option for several sarcoma types [25] and the cornerstone of local control of STSs [26], first-line therapy with surgery is often an indicator of severity, driving our observed association. Alternatively, we observed poorer OS for patients receiving chemotherapy, which again is a marker for disease severity (i.e., more advanced or hard-to-treat sarcomas are often treated using chemotherapy). Unfortunately, recent clinical trials have not been able to demonstrate the advantage of newer chemotherapies over traditional agents, including doxorubicin, especially for STSs [27]. This has largely limited the improvements in outcomes among those receiving chemotherapy.

A notable finding in this study was the associations of smoking status on OS. Few reports have identified this association; however, findings have been equivocal. For example, Zham et al. (1992) reported an increased risk of STS deaths in former and current cigarette smokers compared to non-smokers [28], while Gannon et al. (2018) reported decreased distant metastasis-free and progression-free survival but no changes in OS among current smokers with STS [29]. While we did observe that current smoking was associated with worse OS, this association was not represented in all time periods. Additional work is needed to fully explore the impact of smoking status on OS among sarcoma patients.

Interestingly, there were no significant differences in OS according to race/ethnicity. This supports improving access to a comprehensive cancer center for all patients with sarcoma. In general, incidence rates of sarcomas according to race/ethnicity are heterogenous and vary largely according to histological subtypes [30]. For example, GIST and leiomyosarcoma have a higher incidence among non-Hispanic (NH) Blacks compared to NH-Whites [30,31,32]. Studies demonstrating survival disparities for those diagnosed with sarcomas have been mixed. For example, there is evidence that OS for those with STSs is significantly worse for NH-Black, Asian, and Hispanic patients compared to NH-White patients [32,33,34]. However, a more recent study by Patel et al. (2021) using data from SEER demonstrated that survival for those with STSs was worse in NH-Whites compared to other race/ethnicity groups [35]. The authors attributed these differences to NH-White patients tending to have higher-grade tumors compared to other groups [35,36].

This study should be considered with certain limitations. First, generalizability of our study population might be limited as the study population was derived from a single cancer center and was composed primarily of NH-White individuals. Nonetheless, analysis of sarcoma patients at a tertiary cancer center allows for collection of data from a large number of patients with these otherwise rare cancers. Secondly, data extracted by the MCC Cancer Registry are limited to information that is available in patients’ medical records. Additionally, we were not able to incorporate information on molecular subtypes. In the context of these limitations, this work can be seen as supportive of the idea that separating sarcoma diagnoses rather than consolidating them would improve the chances to observe changes in outcome in interventional studies.

Strengths of our study include a large sample size, the inclusion of several histological sarcoma subtypes, and the availability of data collected over three decades. With over 2000 patients evaluated, our study benefitted from a high statistical power for overall cohort evaluations as well as for subgroup analyses stratified by time period or histological subtype. Furthermore, our study cohort encompassed 43 different ICD-O-3 histology and behavior codes which were further grouped into 13 histological subtypes capturing a large variety of disease and allowing additional disease-specific assessments. The availability of data collected over 28 years allowed us to look for temporal changes often not possible in small cohort studies and capture improvements in treatment.

## 5. Conclusions

In conclusion, we reported on poorer OS rates in older adults (≥40 years), current smokers, patients with metastatic cancer, and patients not receiving first-line surgery treatment from a large study of STS and bone sarcoma patients. Fortunately, improvements in subtyping of sarcomas through improved diagnosis and characterization of recurrent genetic changes, including translocations, along with concerted efforts to better define the roles of chemotherapy, radiation, and surgery has improved individual patient care. Appropriate use of targeted therapies, best exemplified in GIST, experiencing better OS in more recent years, and histology-specific approaches inclusive of immunotherapy, have led to patients forgoing cytotoxic therapies for more effective and less toxic therapy. This subtype-specific approach, often termed “splitting”, will hopefully lead to similar analyses demonstrating more improvements in the coming years. Nonetheless, our findings highlighted the need for future research in sarcomas to better understand barriers to survivorship.

## Figures and Tables

**Figure 1 cancers-15-00514-f001:**
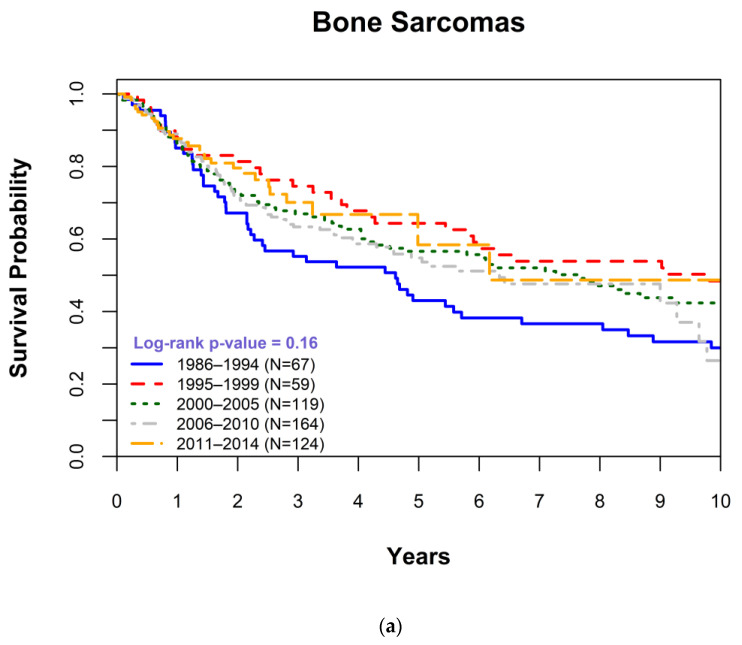
Kaplan-Meier survival plot comparing overall survival (OS) for each time period for patients with (**a**) bone sarcomas, (**b**) soft tissue sarcomas, (**c**) an osteosarcoma, and (**d**) a gastrointestinal stromal tumor. OS was right-censored at 10 years associated and *p*-values were determined using log-rank statistics.

**Table 1 cancers-15-00514-t001:** Demographic, behavioral, and clinical characteristics of individuals diagnosed with sarcoma at MCC, 1986–2014.

Characteristic	N	Unadjusted HR (95% CI)
**Sex**		
Male	1297 (50.5%)	1.00 (referent)
Female	1273 (49.5%)	0.91 (0.82–1.02)
**Age at Diagnosis (yrs)**		
15–39	622 (24.2%)	1.00 (referent)
≥40	1948 (75.8%)	1.28 *** (1.13–1.46)
**Ethnicity**		
Non-Hispanic	2296 (89.3%)	1.00 (referent)
Hispanic	256 (10%)	0.92 (0.76–1.12)
Unknown	18 (0.7%)	1.65 (0.88–3.07)
**Race**		
White	2229 (86.7%)	1.00 (referent)
Black	234 (9.1%)	1.18 (0.98–1.42)
Asian	33 (1.3%)	0.76 (0.41–1.43)
Other	62 (2.4%)	1.06 (0.71–1.58)
Unknown	12 (0.5%)	1.47 (0.70–3.09)
**Tobacco Use**		
Never	1320 (51.4%)	1.00 (referent)
Former User	711 (27.7%)	1.10 (0.96–1.25)
Current User	398 (15.5%)	1.24 ** (1.06–1.44)
Unknown	141 (5.5%)	1.07 (0.85–1.35)
**Sarcoma type**		
Bone Sarcoma	533 (20.7%)	1.00 (referent)
STS	2037 (79.3%)	1.11 (0.97–1.27)
**Spread**		
Localized	1873 (72.9%)	1.00 (referent)
Metastatic	672 (26.1%)	2.45 **** (2.19–2.74)
Undefined	10 (0.4%)	1.24 (0.56–2.78)
Unknown	15 (0.6%)	1.16 (0.60–2.24)
**First Treatment**		
None	88 (3.4%)	1.00 (referent)
Chemotherapy	191 (7.4%)	1.21 (0.87–1.69)
Radiation Therapy	40 (1.6%)	2.20 *** (1.40–3.47)
Surgery	908 (35.3%)	0.39 **** (0.29–0.52)
Multiple Treatments	1343 (52.3%)	0.52 **** (0.39–0.70)
**First Treatment Included Chemotherapy**		
Yes	1047 (40.7%)	1.00 (referent)
No	1523 (59.3%)	0.62 **** (0.56–0.70)
**First Treatment Included Surgery**		
Yes	2167 (84.3%)	1.00 (referent)
No	403 (15.7%)	2.73 **** (2.39–3.13)
**First Treatment Included Radiation**		
Yes	835 (32.5%)	1.00 (referent)
No	1735 (67.5%)	1.06 (0.94–1.19)
**Family history**		
No	1284 (50%)	1.00 (referent)
Yes, one member only	639 (24.9%)	1.03 (0.90–1.17)
Yes, multiple members	647 (25.2%)	0.99 (0.86–1.13)

Abbreviations: STS = soft tissue sarcoma; HR = hazard ratio; CI = confidence interval. ** *p* ≤ 0.01, *** *p* ≤ 0.001, **** *p* ≤ 0.0001.

**Table 2 cancers-15-00514-t002:** Selected demographic and clinical characteristics analyzed in a multivariable model on overall survival in individuals diagnosed with sarcoma at MCC, 1986–2014.

	Overall Survival aHR (95% CI) ^a^
**Age at Diagnosis (yrs)**	
15–39	1.00 (referent)
≥40	1.54 **** (1.34–1.78)
**Ethnicity**	
Non-Hispanic	1.00 (referent)
Hispanic	1.01 (0.83–1.23)
Unknown	2.00 * (1.07–3.74)
**Tobacco Use**	
Never	1.00 (referent)
Former User	1.07 (0.93–1.22)
Current User	1.18 * (1.01–1.37)
Unknown	1.00 (0.79–1.26)
**Sarcoma Type**	
Bone sarcoma	1.00 (referent)
STS	1.07 (0.92–1.23)
**Spread**	
Localized	1.00 (referent)
Metastatic	2.19 **** (1.95–2.47)
Undefined	1.10 (0.49–2.46)
Unknown	1.12 (0.58–2.16)
**First Course of Treatment Included Chemotherapy**	
Yes	1.00 (referent)
No	0.77 **** (0.69–0.87)
**First Course of Treatment Included Surgery**	
Yes	1.00 (referent)
No	2.11 **** (1.82–2.45)
**First Course of Treatment Included Radiation**	
Yes	1.00 (referent)
No	0.97 (0.86–1.09)

Abbreviations: STS = soft tissue sarcoma; aHR = adjusted hazard ratio; CI = confidence interval. ^a^ All characteristics were included in a single multivariable model to calculate aHR. * *p* ≤ 0.05, **** *p* ≤ 0.0001.

**Table 3 cancers-15-00514-t003:** Selected demographic and clinical characteristics in a multivariable model on overall survival in individuals diagnosed with sarcoma at MCC stratified by time period, 1986–2014.

	1986–1994 aHR ^a^ (95% CI)	1995–1999 aHR ^a^ (95% CI)	2000–2005 aHR ^a^ (95% CI)	2006–2010 aHR ^a^ (95% CI)	2011–2014 aHR ^a^ (95% CI)
	N = 207	N = 208	N = 636	N = 795	N = 724
**Age at Diagnosis (yrs)**					
15–39	1.00 (referent)	1.00 (referent)	1.00 (referent)	1.00 (referent)	1.00 (referent)
≥40	2.03 *** (1.37–3.03)	1.72 * (1.10–2.71)	1.29 * (1.02–1.64)	1.35 * (1.03–1.78)	2.21 *** (1.38–3.53)
**Tobacco use**					
Never	1.00 (referent)	1.00 (referent)	1.00 (referent)	1.00 (referent)	1.00 (referent)
Former user	0.80 (0.52–1.24)	1.25 (0.81–1.93)	1.21 (0.96–1.53)	0.97 (0.76–1.22)	1.36 (0.96–1.92)
Current user	0.84 (0.57–1.25)	1.02 (0.62–1.68)	1.39 * (1.06–1.83)	1.05 (0.79–1.39)	1.51 (0.96–2.37)
Unknown	0.72 (0.42–1.23)	0.75 (0.43–1.32)	1.40 (0.92–2.14)	2.19 * (1.02–4.71)	0.96 (0.54–1.72)
**Sarcoma type**					
Bone sarcoma	1.00 (referent)	1.00 (referent)	1.00 (referent)	1.00 (referent)	1.00 (referent)
STS	1.31 (0.88–1.94)	1.69 * (1.11–2.58)	1.19 (0.91–1.56)	0.81 (0.62–1.05)	0.73 (0.47–1.14)
**Spread**					
Localized	1.00 (referent)	1.00 (referent)	1.00 (referent)	1.00 (referent)	1.00 (referent)
Metastatic	2.20 **** (1.58–3.06)	2.48 **** (1.71–3.60)	2.14 **** (1.73–2.64)	2.26 **** (1.82–2.79)	1.86 *** (1.31–2.63)
Undefined	NE	NE	1.52 (0.45–5.10)	0.97 (0.13–6.97)	2.45 (0.59–10.19)
Unknown	0.62 (0.08–4.66)	1.51 (0.54–4.25)	0.49 (0.07–3.56)	NE	1.96 (0.59–6.53)
**First Course of Treatment Included Chemotherapy**					
Yes	1.00 (referent)	1.00 (referent)	1.00 (referent)	1.00 (referent)	1.00 (referent)
No	1.03 (0.72–1.46)	0.74 (0.51–1.09)	0.72 ** (0.59–0.89)	0.66 *** (0.53–0.83)	0.69 * (0.49–0.97)
**First Course of Treatment Included Surgery**					
Yes	1.00 (referent)	1.00 (referent)	1.00 (referent)	1.00 (referent)	1.00 (referent)
No	1.68 * (1.10–2.55)	4.91 **** (3.01–8.01)	2.54 **** (1.95–3.31)	1.72 *** (1.30–2.28)	2.48 **** (1.71–3.59)

Abbreviations: STS = soft tissue sarcoma; aHR = adjusted hazard ratio; CI = confidence interval; NE = not estimated. ^a^ All characteristics were included in a single multivariable model to calculate aHR. * *p* ≤ 0.05, ** *p* ≤ 0.01, *** *p* ≤ 0.001, **** *p* ≤ 0.0001.

## Data Availability

The data that support the findings of this study are available on request from the corresponding author, P.J.L. The data are not publicly available due to their containing information that could compromise the privacy of research participants.

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
