# Peer review of "Trends in Overall Survival among Patients Treated for Sarcoma at a Large Tertiary Cancer Center between 1986 and 2014"

_cancers, 2023, doi:10.3390/cancers15020514_

Round 1

Reviewer 1 Report

Thank you for authors for this interesting paper. 

However, there are some points to correct/clarify. Some of them are major problems with this paper.

Simple summary says that´<40-years old patients were analyzed, but later it says that >40-year old patients had poorer survival. This is controversial, please correct.

It is a common knowledge that later stage and non-operated patients have poorer survival than local and operated ones. Please reformulate this sentence.

GISTs had their first effective treatment (imatinib) in the beginning of 2000. It is obvious that their prognosis has improved since that. The treatment of other soft tissue sarcomas have not changed so much. These two entities should not be analyzed together. They are very different sarcoma entities.

Line 126 and 142 and table 1-3, please differentiate GISTs from other STS.

Please define first treatment more accurately. If patient has as a neoadjuvant therapy radiation therapy, then surgery, is first treatment then only radiation therapy? This is unclear from the paper.

The text in the Table 1 is unclear, as the patients described in the table are not divided according to the vital status (death or alive, as I understand the wording).

Please add aHR to abbreviations below the tables 2 and 3, supplemental table 3.

Please add the explanation why the GIST (Figure 1d) has only records from 2001 onwards. It is not clear for all readers. The most important improvement in the GIST treatment has happened in the first reported years, so it is pitty, if there is no previous data of those patients.

You should discuss the major improvements in sarcoma treatments during the study years (timetable of those, etc) and show if they are seen in this population based data.

Is the patient material in your center selected or are all sarcoma patients in your area treated in your hospital?

Line 249: Please clarify this: Did you include only these 13 histologies in the study or didn´t you have any other histologies during the study period? Otherwise in the text it is understood that every patient was included. In the supplemental tables 2-3 the group of other is 127 patients. The group other should be clarified into the text shortly, even it is in the supplemental table 1.

Please describe your area, the population, geography, etc. to understand the patient material and possible bias (other than ethnicity).

The table 3: it would be more informative if for example adolescents diagnosed in 1986-1994 would be referent to other years and to adult patients. Now you cannot see the change during years in adolescents. Also the title of the table indicates that time is meaningful, but the information does not fulfil this issue. This is the same in every subtitle in the table, only one referent/subtitle is appreciated.

Reviewer 2 Report

The authors have presented a retrospective review of patients with soft-tissue sarcoma and bone sarcoma presenting at a large tertiary care center. I have made some comments in the text of the manuscript that I am uploading. 

One of the major issues with the manuscript is that all STS have been clubbed together. Where there is some advancement in treating particular sarcoma subsets, multiple other subsets continue to have a poor prognosis.  I was expecting to see more granular data, particularly the distribution of head and neck, retroperitoneal, and extremity sarcoma. Unfortunately, that is not the case. 

Methodologically the study is sound. Minor spell checks are needed. 

Round 2

Reviewer 1 Report

Thank you for the corrections.

I would still suggest, that Line 118 "first course of treatment" is clarified. It is not clearly said that all of the given treatments were handled as a "single unit consisting of multiple treatment modalities". Only when you look at the tables this becomes clearer, but it is not enough.

In the supplementary table 2 is still mentioned vital status, as this seems to be a typo, it should be removed.

I did not see the new supplementary table 3 in the provided material.

"Analyzing GIST separately up front is a reordering of the process but does not substantially affect the results" should be discussed also in the text.

Reviewer 2 Report

Thank you for making the edits. 

Author Response

We thank the reviewer for the constructive criticisms and suggestions and have addressed all comments in round 1.